# Coregulators Reside within *Drosophila* Ecdysone-Inducible Loci before and after Ecdysone Treatment

**DOI:** 10.3390/ijms241411844

**Published:** 2023-07-24

**Authors:** Aleksey N. Krasnov, Aleksandra A. Evdokimova, Marina Yu Mazina, Maksim Erokhin, Darya Chetverina, Nadezhda E. Vorobyeva

**Affiliations:** 1Institute of Gene Biology, Russian Academy of Sciences, 119334 Moscow, Russia; krasnov@genebiology.ru (A.N.K.); sashkaa.eevd@gmail.com (A.A.E.); magadovam@yandex.com (M.Y.M.); yermaxbio@yandex.ru (M.E.); daria.chetverina@gmail.com (D.C.); 2Center for Precision Genome Editing and Genetic Technologies for Biomedicine, Institute of Gene Biology, Russian Academy of Sciences, 119334 Moscow, Russia

**Keywords:** transcription, RNA polymerase II, Pol II pausing, Pol II stalling, *Drosophila*, development, transcription regulation, coregulator, ecdysone, 20-hydroxyecdysone, Brm, CBP/p300/Nejire, Rpb3, KisL, CHD1, DART1, dSet1, lid/Kdm5, Gcn5, Spt5, PAF1, cdk8

## Abstract

Ecdysone signaling in *Drosophila* remains a popular model for investigating the mechanisms of steroid action in eukaryotes. The ecdysone receptor EcR can effectively bind ecdysone-response elements with or without the presence of a hormone. For years, EcR enhancers were thought to respond to ecdysone via recruiting coactivator complexes, which replace corepressors and stimulate transcription. However, the exact mechanism of transcription activation by ecdysone remains unclear. Here, we present experimental data on 11 various coregulators at ecdysone-responsive loci of *Drosophila* S2 cells. We describe the regulatory elements where coregulators reside within these loci and assess changes in their binding levels following 20-hydroxyecdysone treatment. In the current study, we detected the presence of some coregulators at the TSSs (active and inactive) and boundaries marked with CP190 rather than enhancers of the ecdysone-responsive loci where EcR binds. We observed minor changes in the coregulators’ binding level. Most were present at inducible loci before and after 20-hydroxyecdysone treatment. Our findings suggest that: (1) coregulators can activate a particular TSS operating from some distal region (which could be an enhancer, boundary regulatory region, or inactive TSS); (2) coregulators are not recruited after 20-hydroxyecdysone treatment to the responsive loci; rather, their functional activity changes (shown as an increase in H3K27 acetylation marks generated by CBP/p300/Nejire acetyltransferase). Taken together, our findings imply that the 20-hydroxyecdysone signal enhances the functional activity of coregulators rather than promoting their binding to regulatory regions during the ecdysone response.

## 1. Introduction

For many years, mechanisms of transcriptional activation by the *Drosophila* 20-hydroxyecdysone (further referred to as ecdysone) hormone have been a subject of interest to many researchers. Ecdysone signaling is the best described and simplest system of steroid action; therefore, it is an ideal experimental model.

Detailed studies by Ashburner’s group on the polytene chromosomes of the *Drosophila* salivary glands corroborated the presence of the ecdysone receptor (EcR) in the nucleus, even in the absence of ecdysone [1,2]. The salivary gland’s response to cultivation in an ecdysone-containing medium was rapid: sensitive loci began to form puffs 10 min after hormone treatment. Ecdysone-dependent regulatory elements must be in a state of high awareness to produce such a rapid response, which requires the constant presence of the EcR.

Since EcR is stably bound to its regulatory elements, the idea of transcription activation by changing the number of coregulators associated with the ecdysone receptor has been suggested. According to this hypothesis, the increase in the ecdysone titer leads to a decrease in the level of corepressors at the enhancers and their association with a set of coactivators, resulting in promoter stimulation and activated transcription [3,4,5].

Indeed, the activation of some ecdysone-inducible model genes is associated with the release of corepressors and the loci’s association with coactivators (such data were obtained for the corepressor SMR and the coactivators TRR and Ash2) [6,7,8]. However, no genome-wide studies have localized various coregulators in ecdysone-responsive loci at a high resolution or studied changes in their binding level during activation. The only exception is an article from the laboratory of A. Brehm, which described the presence of the corepressor Mi-2 in the regulatory regions of the ecdysone-inducible genes and its role as a factor constraining transcriptional overactivation [9].

Previously, our group performed a screening study to characterize the role of 20 different *Drosophila* coregulators in the transcriptional activation of ecdysone-dependent genes [10]. In this research, we used the model genes *dhr3* and *hr4*, which activated in *Drosophila* S2 cells within 20 min of 20-hydroxyecdysone treatment [11]. We identified the most functionally significant coregulators, whose presence in cells was essential for the model genes to reach full activation levels (these were CBP/p300/Nejire, SWI/SNF, Mediator, PAF, and FACT). In addition, we measured changes in the binding level of 20 coregulators to the promoter regions of the model genes within 20 min after 20-hydroxyecdysone treatment. To our surprise, we found a significant level of binding for most of them in the inactive state of genes. Furthermore, the 20-hydroxyecdysone treatment led to no more than a twofold increase in their binding level. Our results questioned the applicability of the transcription activation model by the recruitment of coactivators. However, only the binding level of coregulators to the promoters of model genes was examined, while other regulatory elements of ecdysone-responsive loci were ignored. Here, we used the ChIP-Seq method to study changes in the binding level of coregulators to different regulatory regions of multiple ecdysone-responsive loci.

In addition to promoters and enhancers, architectural protein binding sites were considered in this work as regulatory regions of interest involved in the ecdysone response. It has long been shown that the *Drosophila* architectural proteins Su(Hw) and CP190 are functionally involved in ecdysone responses as positive or negative regulators of ecdysone-dependent transcription, depending on the chromatin context [12]. We recently discovered that the architectural protein CP190 could physically interact with the ecdysone receptor, forming 3D contacts of the CP190 binding sites with promoters and enhancers of genes induced in S2 cells by 20-hydroxyecdysone [13]. According to other data, the ecdysone receptor could interact with a number of other architectural proteins and nuclear pore proteins [14]. We believe that studying the role of architectural proteins binding sites in the ecdysone response deserves special interest.

The term transcriptional coregulator usually refers to proteins and complexes that covalently modify chromatin and remodel the position of nucleosomes. In the class of coregulators, we also tend to include proteins that facilitate the transition of transcription from initiation to the elongation stage (and call them elongators) [15].

The SWI/SNF chromatin-remodeling complex (its ATPase subunit Brm) and acetyltransferase CBP/p300/Nejire, which introduces the H3K27Ac marker of active enhancers, had the strongest functional effect on activating by 20-hydroxyecdysone model *dhr3* and *hr4* genes in S2 cells [10]. The SWI/SNF complex has been found to be critical for transcribing ecdysone-dependent genes during metamorphosis [16]. According to direct and circumstantial data, the ecdysone receptor interacts and functions with the BAP subtype of the SWI/SNF complex (including the specific OSA subunit) [13,17,18]. Cooperative work between CBP/p300/Nejire, SWI/SNF, and the ecdysone receptor in neurons is essential for axon pruning during metamorphosis [17]. An ecdysone-dependent interaction between the EcR-B1 isoform and CBP/p300/Nejire has been demonstrated in co-immunoprecipitation [17]. The recruitment of CBP/p300/Nejire by EcR-B1 to the regulatory elements of the *sox14* gene causes the modification of the chromatin by H3K27Ac and transcriptional activation. CBP/p300/Nejire is also responsible for the acetylation of histone H3 at position 23 in the chromatin of the ecdysone-responsive *eip75b* and *eip74ef* loci [19]. Recently, CBP/p300/Nejire activity on ecdysone-dependent enhancers has been shown in silkworms [20].

From the class of chromatin remodelers, we studied the coregulators CHD1 and Kismet (Kis) in the current work. CHD1 has been known to negatively affect ecdysone-dependent transcription (similar to another member of the CHD family, Mi-2) [21]. Kismet plays a positive role by contributing to the important function of ecdysone or axon pruning during metamorphosis, which is necessary for building new neuronal contacts in the adult brain [22].

Among the classes of coregulators covalently modifying chromatin, we studied DART1/PRMT1, dSet1, lid/Kdm5, and Gcn5. For DART1/PRMT1, an ecdysone-dependent interaction with the ecdysone receptor was shown in coimmunoprecipitation [23]. In our previous experiments, DART1/PRMT1 demonstrated a positive role in the activation of model genes by 20-hydroxyecdysone in S2 cells [10]. At the same time, this protein had a minor but significant negative effect on ecdysone-dependent transcription in earlier experiments using ecdysone-response elements (EcRE) in a transgenic construct [23,24]. Considering the involvement of mammalian PRMT1 in both negative and positive pathways of transcriptional regulation, we believe that *Drosophila* DART1/PRMT1 can influence the transcriptional process in different ways depending on specific contexts [25,26,27]. The coregulators dSet1 and lid/Kdm5 are the main *Drosophila* proteins controlling the balance of H3K4me3 modifications on the promoters of actively working genes [28,29]. Gcn5 acetyltransferase is a component of the SAGA complex, which introduces several histone modifications that positively correlate with gene activity [30].

Among the coregulators controlling the transition from the initiation to the elongation stage of transcription, we analyzed cdk8, PAF, and Spt5 [31,32]. They have shown positive involvement in the ecdysone-induced transcription of model genes in S2 cells [10]. The cdk8 module (part of the Mediator complex, which can function separately) regulates *Drosophila* metabolism by ecdysone [33]. Unlike the core part of the Mediator complex, the cdk8 module interacts with elongating RNA polymerase II and positively contributes to this transcription stage [32,34]. Previously, for Spt5, we showed NELF-dependent recruitment to promoters of the ecdysone-inducible loci in S2 cells [35].

## 2. Results

### 2.1. One-Hour Treatment by Ecdysone Activates 146 Transcripts in Drosophila S2 Cells

Previously, we reported RNA-Seq experiments describing a set of genes with a primary response to 20-hydroxyecdysone treatment in *Drosophila* S2 cells [11,35]. The main idea of the study was to employ a short 1 h treatment, which helped avoid a secondary activation wave arranged by transcriptional factors induced by 20-hydroxyecdysone. Here, we decided to exploit previously published data and make some changes to the analysis. Firstly, we selected particular transcripts whose transcription was increased after the 1 h treatment with 20-hydroxyecdysone by more than 1.4 times. Previously, we reported genes that activated more than twice. This step led to the identification of 146 transcripts depending on 20-hydroxyecdysone (Appendix A). Among them, we selected only transcripts whose genomic loci had EcR peaks in S2 cells (we previously reported FLAG-EcR ChIP-Seq and FLAG-EcR peaks) [13]. The final set used for subsequent analysis in the current article comprised 86 transcripts, which we considered primary responses to 20-hydroxyecdysone in S2 cells (Appendix A).

### 2.2. Ecdysone-Responsive Loci Demonstrated a Moderate Increase in CBP/p300/Nejire Binding at the EcR Sites but a Substantial Increase in H3K27Ac Levels

Our goal for the current study was to describe changes in coregulators’ binding profiles in the regulatory regions of the loci that responded to 20-hydroxyecdysone treatment. To select regulatory regions inside ecdysone-inducible loci in S2 cells, we intersected the peaks of previously reported FLAG-EcR ChIP-Seq and the genomic loci of 86 transcripts induced by 20-hydroxyecdysone. We included +/−5 kb regions encompassing the loci but excluded all TSS regions. We obtained 284 FLAG-EcR-bound peaks inside ecdysone-inducible loci, which we consider primary ecdysone-responsive enhancers in S2 cells (Appendix A). Additionally, we analyzed a set of peaks selected in an alternative way. These peaks were described as ecdysone-inducible enhancers in the A. Stark lab’s STARR-Seq analysis [36]. By contrast, we used a set that only included STARR-Seq enhancers found in our primary ecdysone-responsive loci. Appendix A contains 58 selected STARR-Seq peaks.

According to our previous report, which described RNA polymerase II pausing at the ecdysone-inducible promoters, we detected the presence of Rpb3 at the TSSs of ecdysone-inducible genes before and after a 1 h 20-hydroxyecdysone treatment (Figure 1A) [10,35]. We also observed CBP/p300/Nejire acetyltransferase binding at the TSS before 20-hydroxyecdysone treatment and did not detect an increase in its binding level afterward.

EcR peaks (EcR-bound enhancers) demonstrated an increase in the CBP/p300/Nejire binding level, but it did not exceed two times (Figure 1B). Conversely, the increase in H3K27Ac marks (incorporated by CBP/p300/Nejire) at the EcR peaks after induction was substantial.

Compared with EcR peaks, STARR-Seq peaks demonstrated a substantial increase in the binding levels of FLAG-EcR and CBP/p300/Nejire after 20-hydroxyecdysone treatment (Appendix A). The nature of the selected enhancers can explain this finding. Using STARR-Seq analysis, we identified de novo regulatory regions of the ecdysone-responsive loci where EcR starts to bind after an increase in the 20-hydroxyecdysone titer.

A close examination of the ChIP-Seq tracks at the individual *eip74ef* locus confirms our conclusions from analyzing the averaged plots (Figure 1C,D). The EcR distribution in the *eip74ef* locus perfectly correlates with CBP/p300/Nejire, demonstrating the primary recruitment of EcR to enhancers. Our selection of primary enhancers for averaged plots based on the EcR peak calling (the lowest track in Figure 1C) was correct, as these peaks showed the presence of both EcR and CBP/p300/Nejire, one of the main enhancer components. EcR and CBP/p300/Nejire were present at the enhancers before and after 20-hydroxyecdysone treatment. We detected a substantial increase in H3K27Ac modification levels at the nucleosomes surrounding enhancers proximal to the TSS of the induced *eip74ef* transcript (marked with black). Interestingly, the highest Rpb3 peak inside the *eip74ef* loci was located at the TSS of the transcript, which was not induced by 20-hydroxyecdysone. Moreover, the uninduced *eip74ef* promoter also had EcR and CBP/p300/Nejire peaks proximal to TSS. The only difference between the chromatin context of the surrounding uninducible TSS and the inducible one was a lower level of H3K27Ac modification.

### 2.3. The Brm, CHD1, and KisL Binding Level Mildly Changes at the Ecdysone-Responsive Loci upon Hormone Treatment

Previously, we reported the functional significance of chromatin remodelers for adequately inducing *dhr3* and *hr4* genes by 20-hydroxyecdysone treatment in S2 cells [10]. However, we did not observe a significant increase in the binding of Brm, ISWI, Kismet, CHD1, and Mi-2 remodelers to the promoters of model genes during their induction [10]. Significant portions of all of the studied remodelers were found present at the TSSs before and after 20-hydroxyecdysone treatment. Here, we employed ChIP-Seq to analyze whether Brm, CHD1, and KisL remodelers change their binding level at regulatory regions of primary ecdysone-responsive loci upon treatment.

Despite detecting the presence of Brm and CHD1 at the TSSs of ecdysone-inducible genes in significant amounts, we did not observe a substantial increase in their binding level after a 1 h 20-hydroxyecdysone treatment of S2 cells (Figure 2A). EcR-bound peaks within ecdysone-inducible loci, which we consider primary ecdysone-dependent enhancers, showed less binding for Brm and CHD1 compared to TSSs but more KisL binding (Figure 2B). KisL binding levels at EcR peaks increased twice upon induction. STARR-Seq enhancers showed even less Brm and CHD1 than the EcR peaks but a significant level of KisL binding, especially after a 1 h 20-hydroxyecdysone treatment of S2 cells (Appendix A).

The remodelers’ profiles at the individual *eip74ef* locus had a better colocalization of Brm and CHD1 with TSSs than EcR-bound peaks/enhancers (Figure 2C). Interestingly, within the *eip74ef* locus, the greatest amount of Brm and CHD1 was associated with the TSS of an uninduced transcript. The KisL binding profile inside the *eip74ef* locus correlated with EcR binding, supporting our conclusion—in other words, KisL’s presence at the EcR-bound enhancers. The peculiarity of CHD1 distribution within *eip74ef* has drawn attention to another type of regulatory region. We observed a high CHD1 peak that did not correspond with any TSSs or EcR-bound enhancers. Previously, we reported the functional and structural input of the architectural protein CP190 in regulating ecdysone-inducible loci [13]. Here, we correlated the binding of CHD1 with CP190 within the *eip74ef* locus and found that an additional CHD1 peak perfectly coincided with a single CP190-reach boundary within *eip74ef* [13]. We suggest that the presence of a boundary within the *eip74ef* locus is necessary for the separate regulation of its two TSSs, the activation of which is separated in development and occurs at different levels of the ecdysone hormone titer. The CP190 boundary probably protects the non-inducible in S2 cells promoter from the positive influence of active enhancers surrounding the inducible promoter [5].

To demonstrate that the *eip74ef* locus is not the exception and that the CHD1 remodeler is present at CP190 boundaries within ecdysone-inducible loci, we generated averaged plots of Brm, CHD1, and KisL binding levels at the CP190 sites (N = 140) within 86 loci induced in S2 cells by 20-hydroxyecdysone (Appendix A).

Our findings show the presence of Brm, CHD1, and KisL remodelers within ecdysone-responsive loci in S2 cells before and after hormone treatment. The binding levels of the remodelers demonstrate only minor changes upon 20-hydroxyecdysone induction. The highest amounts of remodelers were present at the TSSs (Brm and CHD1), EcR-bound enhancers (KisL), and CP190-rich boundaries (CHD1). Significant remodeler enrichment was demonstrated by the TSSs of silent and uninducible transcripts (the largest Brm peak was in the uninducible TSSs of the *eip74ef*).

### 2.4. Chromatin Modifiers, Except Gcn5, Reside at EcR-Bound Enhancers before and after 20-Hydroxyecdysone Treatment, with the DART1/PRMT1 Binding Level Substantially Increasing upon Induction

The most important chromatin modifier involved in the ecdysone response is CBP/p300/Nejire, whose knockdown has the severest effect on the transcriptional activation of *dhr3* and *hr4* model genes in S2 cells [10]. Figure 1 shows the recruitment profiles of CBP/p300/Nejire before and after 20-hydroxyecdysone treatment and changes in its functional activity. These changes led to the accumulation of H3K27Ac modifications in the chromatin surrounding the regulatory regions. Here, we discuss an additional set of chromatin modifiers involved in the ecdysone response.

Previously, we found that DART1/PRMT1 binding levels increased several times at the TSSs of *dhr3* and *hr4* model genes after 20-hydroxyecdysone treatment [10]. ChIP-Seq analysis confirmed the active recruitment of DART1/PRMT1 to TSSs and enhancers of various ecdysone-inducible genes after the S2 cells underwent hormone treatment (Figure 3A,B and Appendix A). However, averaged distribution plots and, especially, DART1/PRMT1 distribution at individual loci showed DART1/PRMT1 accumulation at the surrounding chromatin regions and not at the enhancers where EcR and CBP/p300/Nejire bind. This accumulation pattern resembled an increase in H3K27Ac markers (Figure 3C,D). Even though DART1/PRMT1 may be recruited primarily by EcR, it is ultimately found in chromatin rather than in histone-free regions of EcR-bound enhancers [23].

Acetyltransferase Gcn5 was found at the TSSs and enhancers of ecdysone-responsive genes before and after S2 cells’ treatment with 20-hydroxyecdysone (Figure 3A–C). But the biggest amount of Gcn5 was located at the boundary regions, correlating with CP190 distribution (Figure 3C,D and Appendix A). This fact perfectly supports the idea of architectural proteins being the main recruiters of Gcn5 in the *Drosophila* genome [37,38]. In the context of transcriptional regulation by ecdysone, this may mean a functional involvement of Gcn5 in the ecdysone response through a looping model by modifying the regulatory proteins at the TSSs and enhancers but being present at the boundaries.

We found that the dSet1 distribution across the ecdysone-inducible loci correlated well with EcR: dSet1 was present at the TSSs and enhancers, with a tendency to accumulate the EcR-bound enhancers (Figure 3A–D). The dSet1 level at the enhancers slightly increased after the S2 cells were treated with ecdysone. Interestingly, we found that the dSet1 functional antipode lid/Kdm5 bound to TSSs but not the enhancers of ecdysone-inducible genes. In the *eip74ef* locus, we found the highest lid/Kdm5 peak at the inactive and uninducible TSS (Figure 3C). We believe that its activity maintains the inactive state of this TSS [39].

### 2.5. Spt5, PAF1, and cdk8 Elongators Bind the Promoters of Ecdysone-Responsive Genes before and after 20-Hydroxyecdysone Treatment

According to our previous results, the Spt5 subunit of DSIF was found at TSSs of ecdysone-inducible genes before and after S2 cells were treated with 20-hydroxyecdysone (Figure 4) [35]. We did not detect Spt5 binding levels at EcR-bound enhancers. Presumably, Spt5 recruits to promoters directly by RNA polymerase II binding. In the *eip74ef* locus, we found that Spt5 was present both at induced and uninduced TSSs, indicating that its presence at the promoter is not sufficient to activate transcription.

PAF1 and cdk8 showed a similar binding pattern to ecdysone-dependent regulatory regions. They were present at both TSSs and enhancers and only slightly increased the binding level after induction (Figure 4). Similar to Spt5, PAF1 intensively binds to uninducible TSSs at the *eip74ef* locus, showing possible involvement in its suppression. In addition to TSSs and enhancers, cdk8 demonstrates substantial binding levels at the CP190 boundary of the *eip74ef* locus. Therefore, it is possible to incorporate a new member into the boundary-associated complex of coregulators involved in the ecdysone response.

Transcription is activated with ecdysone by stimulating elongation; however, all studied coregulatory complexes involved in elongation were found to be stably associated with ecdysone-responsive loci. This leaves us with two options: to find an as-of-yet unknown elongation regulator that is recruited to ecdysone-responsive loci upon activation or to accept that ecdysone stimulates not recruitment but changes in the functional activity of already associated complexes, causing transcriptional elongation.

## 3. Discussion

Eukaryotic transcription was believed to be activated through a change in the set of coregulators associated with the activated loci for many years [40]. The role of covalent histone modifications in transcriptional activation, increased or decreased at different times, was relegated to a supporting role in the exchange of transcription complexes between regulatory elements [41]. For some chromatin modifiers, it was found that their main functional activity is directed to other transcriptional regulators and not to histone tails [42]. However, the general view of coregulatory behavior has remained more stable. It was assumed that coactivators were recruited to regulatory elements of the genome upon activation.

In terms of the ecdysone response, several coactivators have been found to form protein–protein interactions with the ecdysone receptor when the ecdysone titer is increased [6,8,43]. However, practically no genome-wide studies have been conducted to detect changes in the binding levels of coregulators to ecdysone-dependent genomic elements. We set two tasks for our study: to determine which regulatory elements of the genome are associated with coregulatory complexes in ecdysone-responsive loci and to study changes in the binding levels of coregulators after transcription activation by ecdysone.

### 3.1. Not All Coregulators Involved in Transcriptional Regulation by Ecdysone Are Associated with EcR-Bound Enhancers; Some Preferentially Bind to TSSs and the Boundaries of Ecdysone-Responsive Loci

Our study describes the distribution of 11 various coregulators at the ecdysone-responsive loci in S2 cells before and after hormone treatment. We found that six of the coregulators were enriched at the EcR-bound enhancers: KisL remodeler; CBP/p300/Nejire, DART1, Gcn5, and dSet1 modifiers; and cdk8, involved in elongation control. Brm, CHD1, lid/Kdm5, Spt5, and PAF1 were preferentially bound to the TSSs of ecdysone-inducible genes. Unexpectedly, a substantial amount of CHD1, dSet1, Gcn5, and cdk8 was associated with the CP190-boundary at the *eip74ef* ecdysone-inducible locus, indicating the functional importance of these regulatory elements. We plotted the average binding levels of coregulators for CP190 boundaries located at ecdysone-inducible loci (Appendix A). These plots show that the *eip74ef* locus is not an exception and that CP190 peaks at ecdysone-responsive loci bound by CHD1, dSet1, Gcn5, and cdk8 coregulators. The CHD1, dSet1, and cdk8 binding levels at the CP190 boundaries are comparable to their levels at TSSs and enhancers. By contrast, Gcn5 substantially exceeds binding levels in other regulatory regions. The presence of Gcn5 at the boundaries of ecdysone-responsive loci is consistent with the general view of this protein’s recruitment mechanism at the boundary sites of the *Drosophila* genome [37]. For the other three proteins, this information is new.

Our team and other researchers previously studied the functional involvement of CP190 in the ecdysone response. Its depletion caused inappropriate induction by ecdysone, which was too low for some genes and too high for others [12,13]. We attributed CP190’s functional input to its role in forming correct 3D chromatin structures in ecdysone-responsive loci [13]. The data presented in the current article contribute to the role of CP190 sites. CP190 boundaries serve as a primary landing site for some coregulators participating in the transcriptional regulation of ecdysone-responsive loci. We assumed that some of the coregulators recruited by CP190 could functionally act at the ecdysone-responsive TSS while remaining bound to the CP190 site and approaching the promoter region through loop formation. In future studies, we can verify this assumption by measuring changes in the binding levels of coregulators or the chromatin modification levels introduced by these coregulators at ecdysone-inducible promoters against CP190 boundary destruction.

In general, the presented data show that the coregulators bind various regulatory regions in ecdysone-responsive loci, confirming the formation of a chromatin hub.

### 3.2. Coregulators Bind Regulatory Regions in Ecdysone-Responsive Loci before and after 20-Hydroxyecdysone Treatment

We previously described the detailed recruitment kinetics of 20 various coregulators to TSSs of ecdysone-inducible *dhr3* and *hr4* genes [10]. Surprisingly, we observed significant binding in all coregulators prior to gene induction by 20-hydroxyecdysone and only slight changes afterward. We also detected a significant increase in the covalent histone modifications of chromatin surrounding the TSSs of *dhr3* and *hr4* genes after ecdysone activation. We proposed a mechanism of a “coregulatory pause”, which, similar to the “RNA polymerase II pause”, anticipates the recruitment of inactive coregulators to the promoters of ecdysone-inducible genes and their functional activation upon an increase in the 20-hydroxyecdysone titer. Nonetheless, there was a possibility that the proposed mechanism operated exceptionally in *dhr3* and *hr4* promoters. In the current manuscript, we present data supporting our previous ideas and suggesting the presence of coregulators at multiple ecdysone-inducible loci prior to transcriptional activation.

All 11 studied coregulators demonstrated considerable binding levels to regulatory regions of ecdysone-responsive loci in their inactive state. DART1/PRMT1 increased multifold after S2 cells were treated with 20-hydroxyecdysone, whereas the rest remained practically unchanged. Despite a significant increase in DART1/PRMT1 binding levels, it is hardly suitable for the role of a coregulator switching the status of ecdysone-dependent genes based on available functional data. In our early studies, its knockdown had a relatively weak effect on the ecdysone activation of the *dhr3* and *hr4* model genes [10]. Upon studying the response to ecdysone using a transgenic reporter construct carrying ecdysone-response elements (EcRE), DART1/PRMT1 worked as a repressor of ecdysone-dependent induction [23]. Determining the exact functional contribution of DART1/PRMT1 to ecdysone-dependent induction requires further research. So far, we assume that its functional role may be more closely related to the mRNA processing of ecdysone-responsive genes than to the function of a direct RNA polymerase II regulator [44,45].

For most of the coregulators (except DART1/PRMT1), we observed mild changes in binding levels after 20-hydroxyecdysone treatment. Nevertheless, we admit that these minor changes may carry functional significance. In the case of the ecdysone response, mild changes in the coregulators’ binding levels affect several enhancers that form the clusters controlling the transcriptional status of individual ecdysone-dependent genes, as described previously [36]. Synchronous changes in several regulatory elements, along with minor changes in the binding levels of many coregulators, can be critical and lead to changes in the transcriptional status of the gene. Moreover, we must consider the limitations of the ChIP-Seq technique, which only detects proteins directly associated with chromatin. Using this method, we cannot analyze changes in coregulator concentrations in the nucleoplasm surrounding the regulated locus, which can be increased via phase separation mechanisms [46]. In this case, a phase condensate enriched with coregulators can form around the regulated locus, and the functional activity of coregulators in regulatory elements can increase according to the “hit-and-run” mechanism, leaving coregulators’ binding level to chromatin unchanged.

It is worth noting that we detected a substantial increase in H3K27Ac modification at the EcR-bound enhancers upon induction with 20-hydroxyecdysone. Considering that CBP/p300/Nejire is stably present at the EcR-bound enhancers, this finding suggests CBP/p300/Nejire recruitment in a functionally inactive state (or “paused state”). Our data suggest that an increase in the ecdysone titer can stimulate CBP/p300/Nejire activity on regulatory elements of the genome. For now, we have considered three ways of implementing this. First, it is known that human CtBP can directly interact with p300 and inhibit its acetyltransferase activity [47,48]. We believe that CBP/p300/Nejire can bind in the form of a complex with CtBP to the inactive ecdysone-responsive *Drosophila* loci. In this case, exposure to ecdysone may disrupt this interaction by removing CtBP and, consequently, stimulate CBP/p300/Nejire acetyltransferase activity. An alternative possibility is the simultaneous binding of inactive ecdysone-responsive loci by enzymes with opposing activities regarding H3K27Ac modification, namely, CBP/p300/Nejire acetyltransferase and Rpd3/HDAC1 deacetylase (a previously described repressor of ecdysone-inducible genes) [49,50]. In this case, ecdysone may increase the level of H3K27Ac by promoting the dissociation of Rpd3/HDAC1 from regulatory elements. The third option is to stimulate CBP/p300/Nejire activity with ecdysone by activating eRNA synthesis at the enhancers (eRNA is known to interact with CBP/p300 and aggravate its activity [51]) or covalent modification of the enzyme itself by recruiting a not-yet-found *Drosophila* modifier [52,53].

Chromatin binding by coregulators in a functionally inactive state has found indirect confirmation in studies that split chromatin into various functional subtypes (chromatin colors). The first studies aimed at dividing genome regions into functional subtypes and relied more on the state of histones and their modifications than on the presence of particular coregulatory proteins [54]. This approach contributed significantly to the field but did not lead to the isolation of loci, containing genes with similar regulatory mechanisms. More recent studies, which sort chromatin based on the transcriptional regulators distribution rather than modifications, made it possible to link the chromatin state of the regions with their functional properties—for example, maintaining the transcription of housekeeping or tissue-specific genes [55,56]. This information suggests that the presence of a coregulator on chromatin may not necessarily mean its active functional state. This hypothesis is supported, for instance, by a recent study that assigned distinct active/repressed states to different types of chromatins (separate SWI/SNF and SWI/SNF-R, as well as yellow and yellow-R chromatins, distinguished between active/repressed developmental and active/repressed housekeeping genes, correspondingly) [55]. In the context of new data, ecdysone-dependent transcriptional regulation suggests an inactive-to-activated-state transition, with slight changes in the coregulatory complex composition associated with ecdysone-responsive loci but also with changes in their functional activity (realized by the remodeling and modifications of histones and transcriptional factors). Details concerning these changes have yet to be defined.

## 4. Materials and Methods

### 4.1. Treatment of Drosophila S2 Cells

*Drosophila* Schneider cell line 2 (S2) cells were maintained at 25 °C in ecdysone-free Schneider’s insect medium (S9895, Sigma-Aldrich, St. Louis, MO, USA) containing 10% FBS (CH30160.03, HyClone, Erembodegem, Belgium). The treatment of S2 cells with 20-hydroxyecdysone (20E) (H5142, Sigma-Aldrich, St. Louis, MO, USA) was performed at a final concentration of 0.3 μM, as was described previously [10,11]. Our approach was aimed at identifying only direct targets of 20E by limiting the time of 20E treatment to 1 h. This prevents primary-induced transcripts (which are, in turn, transcription factors) from being translated and shuttled back to the nucleus to affect the transcription of their targets.

### 4.2. ChIP-Seq Analysis

The chromatin immunoprecipitation (ChIP) and ChIP-MNase seq were performed and analyzed exactly as previously described [10,35,57]. ChIP-Seq libraries were obtained using the NEBNext UltraTM II DNA library preparation kit (E7645S, New England Biolabs, Ipswich, MA, USA). Only the library fragments of 350–500 bp were subjected to NGS sequencing. Next-generation sequencing was performed by Evrogen (evrogen.ru) with the Illumina NovaSeq6000 sequencer. For each of the ChIP-Seq libraries, approximately 8–10 million unique paired-end reads were obtained. The paired-end reads in FastQ format were mapped to the *Drosophila* genome assembly dm6 using HISAT2 [58] and filtered (with a minimum MAPQ quality score = 10). The Deeptool2 package was used for the further analysis of the obtained data [59]. BigWig files were generated using bamCoverage 3.0.2 with scores representing the number of reads normalized by the size of the library (the protein binding levels were normalized to the genome content—calculated as RPGC: number of reads per bin/(total number of mapped reads * fragment length/effective genome size)) [59]. The final bigwig files (representing the protein binding profiles) were obtained using the bigwigcompare tool as a ratio of the ChIP signal to the input (all inputs were preliminary smoothed over a 1 kb window). Pile-up profiles were calculated as a mean level of protein binding. The background binding level for the ChIP-Seqs was estimated using the set of 500 randomly chosen regions of the *Drosophila* genome and was provided as a grey transparent area at the averaged plots.

A set of primary transcripts induced in S2 *Drosophila* cells after 1 h of treatment with 20-hydroxyecdysone was selected using previously described RNA-Seq data (GEO Series GSE156847) [10,11]. First, we selected particular transcripts of genes whose transcription was increased after a 1 h treatment with ecdysone by more than 1.4 times (previously, we reported genes increased by more than two times). This led to the identification of 146 transcripts that depend on ecdysone in S2 *Drosophila* cells. Next, we selected only transcripts whose genomic loci have EcR peaks in S2 untreated cells (FLAG-EcR ChIP-Seq and FLAG-EcR peaks were previously reported by our group) [13]. The resulting set comprised 86 transcripts which we consider as primary respondents to ecdysone in S2 cells (provided as a bed in the Appendix A).

To pick the enhancers inside primary ecdysone-inducible loci in S2 cells, we intersected the peaks of previously reported FLAG-EcR ChIP-Seq (we merged sets of peaks called for the cells treated for 1 h with 20-hydroxyecdysone and the sham-treated S2 cells) and genomic loci of 86 transcripts induced by 20-hydroxyecdysone. We included +/−5 kb regions encompassing the loci but excluded all TSS regions (as genomic loci of the transcript, we take regions of the genome where both induced and non-induced transcripts of the current gene are present; this gave us 284 EcR-bound peaks in 20-hydroxyecdysone-responded loci) (provided as a bed in the Appendix A)). We obtained the set of STARR-Seq enhancers by the intersection of a previously published list of regions with the ability to provide ecdysone-dependent transcription induction in *Drosophila* S2 cells with our list of 86 loci that are primarily respondent to ecdysone (as in the case of EcR-peaks, we included +/−5 kb regions encompassing the loci but excluded all TSS regions during intersection; in total, we generated a list of 58 STARR-Seq enhancers that were used in our analysis; provided as a bed in the Appendix A).

We did not use any figures or text from the previously published manuscripts—only data deposited in free-access databases. The Galaxy-P platform was used for the analysis of ChIP-Seq data [60].

All of the processed files (both bigwig and bed) were obtained with coordinates of the *Drosophila* genome assembly dm6.

## 5. Antibodies

Rabbit polyclonal antibodies against Rpb3 (1–275 aa), Brm (1513–1646 aa), CBP/p300/Nejire (125–310 aa), KisL (1150–1314 aa), CHD1 (1420–1651 aa), DART1 (1–100 aa), dSet1 (1–103 aa), lid/Kdm5 (1–165 aa), Gcn5 (349–813 aa), Spt5 (917–1117 aa), PAF1 (1–234 aa), and cdk8 (293–394 aa) were obtained and described previously in our lab [10,35]. All antibodies were affinity-purified. All of them were tested in ChIP experiments using the *hsp70* gene induced by heat shock [61]. Antibodies production was performed according to the procedures outlined in the NIH (USA) Guide for the Care and Use of Laboratory Animals. The protocol used was approved by the Committee on Bioethics of the Institute of Gene Biology of the Russian Academy of Sciences. All procedures were performed under conditions designed to minimize suffering. Antibodies against the histone H3K27Ac (39133) were purchased from Active Motif.

## Figures and Tables

**Figure 1 ijms-24-11844-f001:**
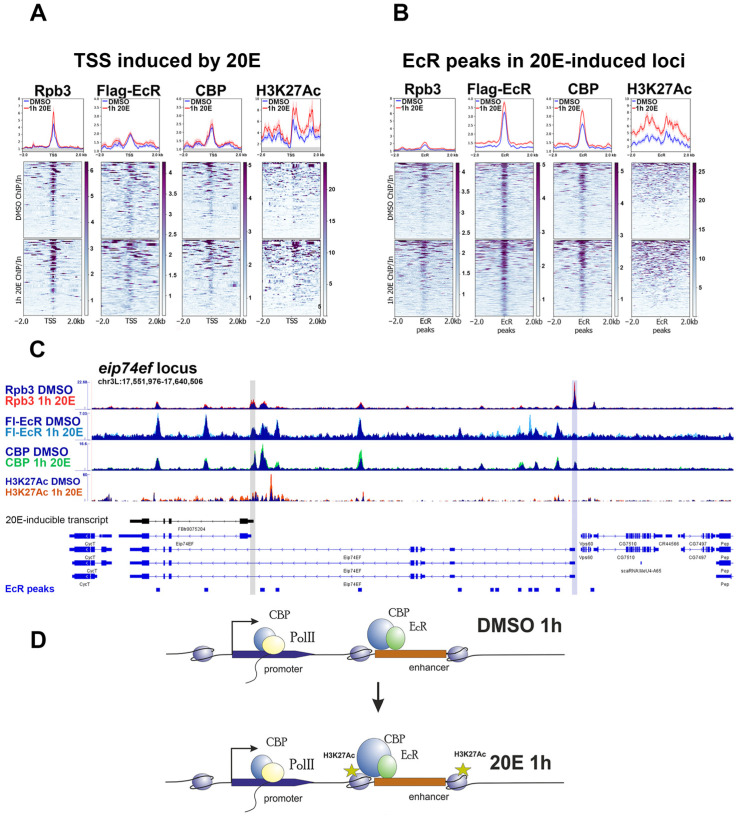
Binding profiles of RNA polymerase II, FLAG-EcR, and CBP/p300/Nejire and the level of H3K27Ac modification at the regulatory regions of the ecdysone-responsive loci in *Drosophila* S2 cells. The figure shows the average distribution of RNA-polymerase II (Rpb3 subunit), FLAG-EcR, CBP/p300/Nejire, and Histone H3K27Ac modification at the TSSs (transcription start sites, N = 86) (**A**) and FLAG-EcR peaks (N = 284) (**B**) of the genomic loci, whose transcription is induced by a 1 h treatment of Drosophila S2 cells with 20-hydroxyecdysone. ChIP-Seqs and ChIP-MNase Seq for H3K27Ac were per-formed on Drosophila S2 cells treated with 20-hydroxyecdysone (0.3 µM) for 1 h “1h 20E” (red graph) or on the sham-treated cells “DMSO” (blue graph). Protein binding levels were calculated as an enrichment (ratio of the corresponding ChIP-Seq signal to the input DNA). Average profiles were calculated as the mean of the protein binding level. The standard error appears on the graphs as a lighter area around the main line of the profiles. The background binding level for the ChIP-Seqs was estimated using the set of 500 randomly chosen regions of the Drosophila genome and was provided as a grey transparent area at the averaged plots. (**C**) The figure shows binding profiles of Rpb3, FLAG-EcR, CBP/p300/Nejire, and H3K27Ac at the eip74ef ecdysone-inducible locus. The grey line indicates an inducible TSS and the blue line indicates a non-inducible one. The protein binding levels were estimated by ChIP-Seq. ChIP-Seq experiments were performed using 1h 20E- or DMSO-treated Drosophila S2 cells. ChIP-Seq data on FLAG-EcR (GSE139316), Rpb3 (GSE102520), and CBP/p300/Nejire (GSE156847) binding were loaded from GEO and described previously. ChIP-MNase-Seq data on H3K27Ac are described here for the first time. (**D**) The figure shows a schematic model describing changes in the protein binding level of RNA polymerase II, FLAG-EcR, CBP/p300/Nejire, and H3K27Ac with regulatory regions of the ecdysone-inducible loci in Drosophila S2 cells before and after their treatment with 20-hydroxyecdysone. The sizes of the depicted coactivators, histone modification, and RNA polymerase II reflect changes in the level of their binding to the regulatory element.

**Figure 2 ijms-24-11844-f002:**
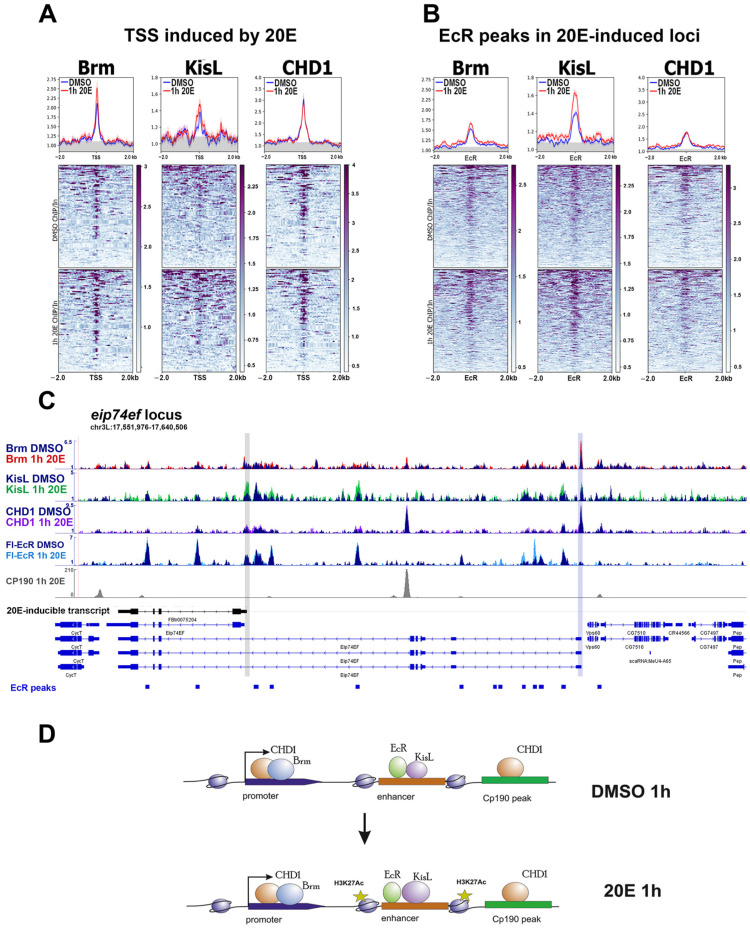
Binding profiles of Brm, KisL, and CHD1 at the regulatory regions of the ecdysone-responsive loci in *Drosophila* S2 cells. The figure shows the average distribution of Brm, KisL, and CHD1 at the TSSs (transcription start sites, N = 86) (**A**) and FLAG-EcR peaks (N = 284) (**B**) of the genomic loci, whose transcription is induced by a 1 h treatment of Drosophila S2 cells with 20-hydroxyecdysone. ChIP-Seqs were performed on Drosophila S2 cells treated with 20-hydroxyecdysone (0.3 µM) for 1 h “1h 20E” (red graph) or on the sham-treated cells “DMSO” (blue graph). The protein binding levels were calculated as an enrichment (ratio of the corresponding ChIP-Seq signal to the input DNA). The average profiles were calculated as the mean of the protein binding level. The standard error appears on the graphs as a lighter area around the main line of the profiles. The background binding level for the ChIP-Seqs was estimated using the set of 500 randomly chosen regions of the Drosophila genome and was provided as a grey transparent area at the averaged plots. (**C**) The figure shows the binding profiles of Brm, KisL, CHD1, FLAG-EcR, and CP190 at the eip74ef ecdysone-inducible locus. The grey line indicates an inducible TSS and the blue line indicates a non-inducible one. The protein binding levels were estimated by ChIP-Seq. ChIP-Seq experiments were performed using 1 h 20E- or DMSO-treated Drosophila S2 cells. ChIP-Seq data on FLAG-EcR (GSE139316) and CP190 (GSE156847) binding were loaded from GEO and described previously. ChIP-Seq data on Brm, KisL, and CHD1 are described here for the first time and provided in GSE102520. (**D**) The figure shows a schematic model describing changes in the protein binding level of Brm, KisL, CHD1, FLAG-EcR, and H3K27Ac with regulatory regions of the ecdysone-inducible loci in Drosophila S2 cells be-fore and after their treatment with 20-hydroxyecdysone. The sizes of the depicted coactivators, histone modification, and RNA polymerase II reflect changes in the level of their binding to the regulatory element.

**Figure 3 ijms-24-11844-f003:**
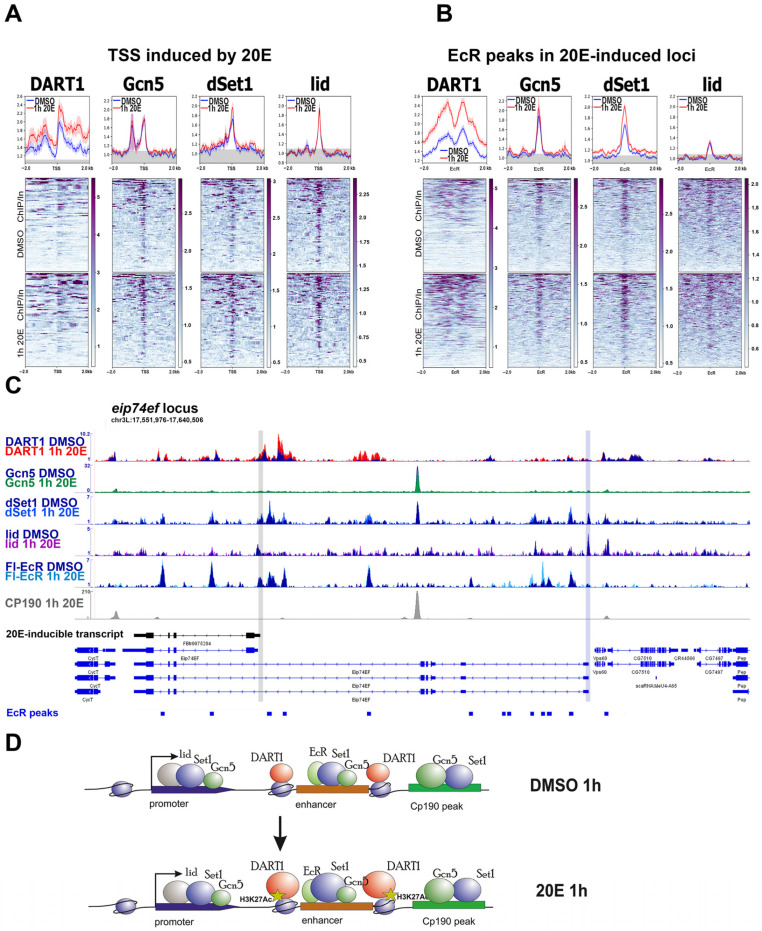
Binding profiles of DART1, Gcn5, dSet1, and lid/Kdm5 at the regulatory regions of the ecdysone-responsive loci in *Drosophila* S2 cells. The figure shows the average distribution of DART1, Gcn5, dSet1, and lid/Kdm5 at the TSSs (transcription start sites, N = 86) (**A**) and FLAG-EcR peaks (N = 284) (**B**) of the genomic loci, whose transcription is induced by a 1 h treatment of Drosophila S2 cells with 20-hydroxyecdysone. ChIP-Seqs were performed on Drosophila S2 cells treated with 20-hydroxyecdysone (0.3 µM) for 1 h “1h 20E” (red graph) or on the sham-treated cells “DMSO” (blue graph). Protein binding levels were calculated as an enrichment (ratio of the corresponding ChIP-Seq signal to the input DNA). Average profiles were calculated as the mean of the protein binding level. The standard error appears on the graphs as a lighter area around the main line of the profiles. The background binding level for the ChIP-Seqs was estimated using the set of 500 randomly chosen regions of the Drosophila genome and was provided as a grey transparent area at the averaged plots. (**C**) The figure shows the binding profiles of DART1, Gcn5, dSet1, lid/Kdm5, FLAG-EcR, and CP190 at the eip74ef ecdysone-inducible locus. The grey line indicates an inducible TSS and the blue line indicates a non-inducible one. The protein binding levels were estimated by ChIP-Seq. ChIP-Seq experiments were performed using 1 h 20E- or DMSO-treated Drosophila S2 cells. ChIP-Seq data on FLAG-EcR (GSE139316) and CP190 (GSE156847) binding were loaded from GEO and described previously. ChIP-Seq data on DART1, Gcn5, dSet1, and lid/Kdm5 are described here for the first time and provided in GSE102520. (**D**) The figure shows a schematic model describing changes in the protein binding level of DART1, Gcn5, dSet1, lid/Kdm5, FLAG-EcR, and H3K27Ac with regulatory regions of the ecdysone-inducible loci in Drosophila S2 cells before and after their treatment with 20-hydroxyecdysone. The sizes of the depicted coactivators, histone modification, and RNA polymerase II reflect changes in the level of their binding to the regulatory element.

**Figure 4 ijms-24-11844-f004:**
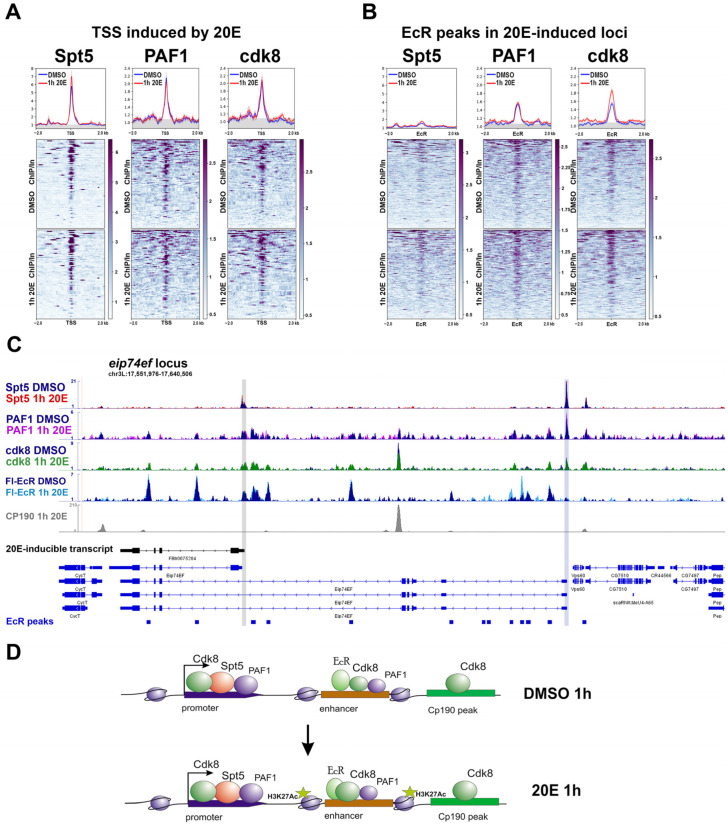
Binding profiles of Spt5, PAF1, and cdk8 at the regulatory regions of the ecdysone-responsive loci in *Drosophila* S2 cells. The figure shows the average distribution of Spt5, PAF1, and cdk8 at the TSSs (transcription start sites, N = 86) (**A**) and FLAG-EcR peaks (N = 284) (**B**) of the genomic loci, whose transcription is induced by a 1 h treatment of Drosophila S2 cells with 20-hydroxyecdysone. ChIP-Seqs were performed on Drosophila S2 cells treated with 20-hydroxyecdysone (0.3 µM) for 1 h “1h 20E” (red graph) or on the sham-treated cells “DMSO” (blue graph). Protein binding levels were calculated as an enrichment (ratio of the corresponding ChIP-Seq signal to the input DNA). Average profiles were calculated as the mean of the protein binding level. The standard error appears on the graphs as a lighter area around the main line of the profiles. The background binding level for the ChIP-Seqs was estimated using the set of 500 randomly chosen regions of the Drosophila genome and was provided as a grey transparent area at the averaged plots. (**C**) The figure shows the binding profiles of Spt5, PAF1, cdk8, FLAG-EcR, and CP190 at the eip74ef ecdysone-inducible locus. The grey line indicates an inducible TSS and the blue line indicates a non-inducible one. The protein binding levels were estimated by ChIP-Seq. ChIP-Seq experiments were performed using 1h 20E- or DMSO-treated Drosophila S2 cells. ChIP-Seq data on FLAG-EcR (GSE139316) and CP190 (GSE156847) binding were loaded from GEO and described previously. ChIP-Seq data on Spt5, PAF1, and cdk8 are described here for the first time and provided in GSE102520. (**D**) The figure shows a schematic model de-scribing changes in the protein binding level of Spt5, PAF1, cdk8, FLAG-EcR, and H3K27Ac with regulatory regions of the ecdysone-inducible loci in Drosophila S2 cells be-fore and after their treatment with 20-hydroxyecdysone. The sizes of the depicted coactivators, histone modification, and RNA polymerase II reflect changes in the level of their binding to the regulatory element.

## Data Availability

All obtained ChIP-Seq data were deposited into the Gene Expression Omnibus—GSE102520.

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
