# Peer review of "Coregulators Reside within Drosophila Ecdysone-Inducible Loci before and after Ecdysone Treatment"

_ijms, 2023, doi:10.3390/ijms241411844_

Round 1
Reviewer 1 Report
Krasnov et al. investigate the regulation of transcription factor/coregulator recruitment in response to ecdysone hormone in Drosophila cells. To do this, they used chromatin immunoprecipitation and sequencing with a focus on gene loci thought to be regulated by ecdysone. While the general model for steroid hormone regulation suggests that co-regulators may bind upon addition of ecdysone, this work indicates that many factors are already bound, and suggests their activity changes in response to the hormone. In particular, H3K27 acetylation is clearly changed when ecdysone is added for many sets of target loci. The results are substantiated for the eip74ef locus. The study also suggests longer-distance effects may be important. In general, the study is comprehensive and a lot of data is presented. At times, the data or figures could be presented/described more clearly. I have some specific comments below.
- The authors make a big deal about their data changing the prevailing model of the field, but I think the current accepted model in the field is closer to their new model - and they provide a larger set of good evidence for this. So the authors do not need to overstate a change to the model.
- Similarly, the authors imply that a less-than 2-fold difference in binding occupancy is too small to have a major effect. However, if multiple different factors have a small change in binding, that could have a bigger effect, and the fold changes, when averaged, may appear smaller. Or there may be a significant temporal component in which the binding is briefly changed. Another caveat is this work is mostly completed in one cell type. So the interpretation of this could be expanded.
- The authors should provide more background on boundaries in the introduction and explain the presence of a boundary in the eip74ef locus.
- Some data in the figures are hard to see because the graphs are very small and the colors are close. The standard errors and the axes labels are especially hard to discern. It would be helpful if the graphs were bigger.
- The schematic models in part E of the figures are helpful, but it should be explained in the text that the bigger ovals mean more binding- it is not obvious.
- Since the authors compare the TSS of the inducible and uninducible eip74ef locus, I think it would be beneficial to show more data to the right of the uninducible locus. I think that is downstream but binding there still could affect the transcription and would make the comparison more clear.
- There are some minor grammatical errors throughout and Nejire is spelled wrong in multiple places.
fine- minor corrections needed
Reviewer 2 Report
The manuscript “Coregulators Reside at the Drosophila Ecdysone-Inducible Loci Both before and after Ecdysone Treatment” by Krasnov et al. Describes a number of ChIP-seq experiments against various transcriptional activator proteins at ecdysone responsive genes in S2 cells. Basically, the authors confirm and expand previous results suggesting that most activator proteins are present at the TSS and enhancer sites prior to ecdysone stimulation, suggesting that ecdysone bound to its receptor does not recruit proteins to the enhancer but instead changes their activity to stimulate transcription. Furthermore, the authors show that many of these activators also bind to areas associated with insulator activity (CP190 binding peaks). Previously, the authors showed that CP190 was required for peak activation of some ecdysone responsive genes. Thus, the authors hypothesize that insulator mediated chromatin looping interactions might play a role in ecdysone-dependent gene activation. Overall, I found the work interesting and valuable. I would recommend publication with some minor cosmetic modifications and some additional discussion elements.
1. Many problems with the writing exist. There are many grammatical errors (ex. Line 126 “...which let to avoid...”, and “ line 139-140 “ changes in coregulators binding profiles at the regulatory regions of the loci respondeding to …, line 129... line 156..., line 175-177... among many others). I would suggest that the authors send the article to a native speaker (scientist) to correct these errors. Alternatively, perhaps ChatGPT might be able to find these errors, and correct them under the supervision of the authors.
2. The figures are way too small and of poor resolution in the document provided. Looking at the manuscript on screen helped a tiny bit, but the resolution still made examination difficult. The text is almost unreadable and examining the trace colors is sometimes impossible. If someone wants to print the article it is even worse (older generation people or people who like to walk around while reading, like me). The figures need to be larger and of better quality. If the journal is being stingy about space, I suggest that you reduce the number of panels somehow to increase figure size of the rest. Put the suppressed figures into a supplementary file.
3. The authors focus on the fact that most coregulators seem to be bound to the ecdysone responsive genes regardless of the presence or absence of ecdysone. The authors do not discuss how the activity of the coregulators could be controlled by ecdysone. What is known about how the enzymatic activity of these coregulators is controlled outside of simple recruitment? Alternatively, do the authors believe other proteins might be recruited to regulate their activity? Have screens pointed to some candidates? Also, there are slight increases in the binding of some proteins. Could these slight changes be significant?
4. One protein was found to be highly enriched after ecdysone treatment. This protein is DART1, a negative regulator of ecdysone receptor activity that is known to associate with ecdysone receptor upon ecdysone stimulation. Could the authors discuss why this molecule would be recruited to ecdysone activated genes while activator proteins are not (with supporting evidence, of course)?
5. While I don’t expect this experiment to be done, I wonder if the authors can try to use a boundary bypass assay “à la” Pavel Georgiev to see if ecdysone might allow distal enhancers to interact with promoters using the eip74ef promoter and insulator.
See above
Reviewer 3 Report
Drosophila is used for many years as model for transcriptional activation by ecdysteroid hormones. Recently, the authors have modified the mechanisms of transcription activation upon ecdysteroids by presenting novel/additional coregulators like CBP, SWI/SNF, Mediator, PAF, and FACT in Drosophila S2 cells. Data in the present manuscript support the idea of many more coregulators at multiple ecdysteroid-inducible loci prior to their transcriptional activation. Some of them are described in their function for the first time. In this respect the manuscript does not show new research approaches, but represents a logical continuation of former research in the group-
My only major concern on the paper is, that authors treated the cells with 20-hydroxaecdysone, but talk about ecdysone treatment in the entire manuscript. Ecdysone and 20-hydroxyecdysone are two ecdysteroids, different in structure!
Minors:
- species names must be written in italics in the entire manuscript (Drosophila also!)
- line 139: to describe
- line 170: are present
- line 174: promoter also possesses
- line 189: binding level was
- line 228: level was estimated
- line 246: an additional peak
-line 381: insert a full stop after the last word
- line 582: Drosophila melanogaster
English needs only minor corrections; see above
